# Dermatological Side Effects of Cancer Treatment: Psychosocial Implications—A Systematic Review of the Literature

**DOI:** 10.3390/healthcare11192621

**Published:** 2023-09-25

**Authors:** Vera Almeida, Daniela Pires, Marta Silva, Maribel Teixeira, Ricardo João Teixeira, André Louro, Maria Alzira Pimenta Dinis, Maria Ferreira, Ana Teixeira

**Affiliations:** 1UNIPRO—Oral Pathology and Rehabilition Research Unit, University Institute of Health Sciences (IUCS), CESPU, CRL, 4585-116 Gandra, Portugal; vera.almeida@iucs.cespu.pt; 2UCIBIO—Applied Molecular Biosciences Unit, MedTech-Laboratory of Pharmaceutical Technology, Faculty of Pharmacy, University of Porto, 4050-313 Porto, Portugal; ana.teixeira@iucs.cespu.pt; 3University Institute of Health Sciences (IUCS), CESPU, CRL, 4585-116 Gandra, Portugal; danielaa.pires@hotmail.com (D.P.); martascoelho4@hotmail.com (M.S.); maria.rferreira@hotmail.com (M.F.); 41H-TOXRUN—One Health Toxicology Research Unit, University Institute of Health Sciences (IUCS), CESPU, CRL, 4585-116 Gandra, Portugal; 5REACH—Mental Health Clinic, 4000-138 Porto, Portugal; louro.andre@gmail.com; 6CINEICC, Faculty of Psychology and Educational Sciences, University of Coimbra, 3004-531 Coimbra, Portugal; 7RECI—Research Unit in Education and Community Intervention, Instituto Piaget—ISEIT/Viseu, 1950-157 Viseu, Portugal; 8UFP Energy, Environment and Health Research Unit (FP-ENAS), University Fernando Pessoa (UFP), Praça 9 de Abril 349, 4249-004 Porto, Portugal; madinis@ufp.edu.pt; 9Fernando Pessoa Research, Innovation and Development Institute (FP-I3ID), University Fernando Pessoa (UFP), Praça 9 de Abril 349, 4249-004 Porto, Portugal

**Keywords:** dermatological side effects, psychosocial implications, alopecia, body image, cancer treatment, cognitive fusion, quality of life, social inhibition

## Abstract

Cancer is a leading cause of mortality and morbidity all over the world and the second major cause of death in Portugal. Dermatological side effects resulting from cancer treatment have a psychosocial impact on patients’ lives, such as quality of life (QoL), body image, cognitive fusion and social inhibition. This systematic review aimed to explore and synthesize the psychosocial impact of dermatological side effects of cancer treatment, answering the following research objectives: (i) Do the dermatological side effects of the cancer treatment present any psychosocial impact for the patients? (ii) How does the psychosocial impact of the dermatological toxicities of the cancer treatment manifest in patients’ lives? Preferred Reporting Items for Systematic Reviews and Meta-Analyses (PRISMA) guidelines were followed and guided a systematic search through the PubMed, Cochrane Library and PyscNet databases. The considered studies correlate dermatological side effects of cancer treatments and their psychological/psychosocial outcomes. The studies found were all published in peer-reviewed journals. The results obtained established that cancer treatment causes the most varied skin changes, consequently reducing self-esteem and QoL; disturbing body image; and contributing to cases of stress, depression and anxiety. There is still limited literature that profoundly investigates the experience of living with these skin toxicities. The development of research lines to improve knowledge in this field will allow for significant improvements in healthcare for patients undergoing cancer treatment who need to focus more on the psychosocial implications of skin toxicities. The novelty of this review lies in adding knowledge summarizing the psychosocial implications of dermatological side effects of cancer treatment to support healthcare providers in the development of integrative therapeutic strategies for these patients in their clinical practice.

## 1. Introduction

Cancer continues to be a leading cause of mortality and morbidity worldwide [1]. Most of the existing oncological diseases are treated based on chemotherapy or radiotherapy alone or in combination with other treatments, with the side effects of these treatments being a clinical limitation of their administration. Health-related side effects induced by the cancer treatment may appear during or a long time after the treatment ends and often rely on the patient’s radiosensitivity [2]. The patient receiving cancer treatment usually has severe side effects such as pain, nausea, diarrhea, cardiotoxicity, depression of the immune system, hair loss, disruption of the cutaneous barrier, and changes in skin color and dryness [3,4,5,6]. This cutaneous toxicity could change the body image of the patient with significant psychosocial impact. Likewise, individuals suffering from skin diseases frequently encounter a situation characterized by physical disfigurement, psychological distress and societal stigma [7]. Dose reduction and discontinuation of treatment are not suitable options to avoid dermatological side effects since they can adversely affect cancer treatment outcomes [8].

The present review aims to synthesize and discuss the evidence from studies examining the psychosocial impact of dermatological effects resulting from cancer treatment and assess whether the psychological and emotional impact is integrated into the therapeutic strategies of the treatment of cancer patients. Thus, this review intends to answer the two following main research questions:

RQ1. Do the dermatological side effects of the cancer treatment present any psychosocial impact on the patients?

RQ2. How does the psychosocial impact of the dermatological toxicities of cancer treatment manifest in patients’ lives?

## 2. Background

Dermatological toxicity is assumed to be a common side effect caused by cancer treatment [9], which, if not properly managed, can become uncomfortable and disfiguring [6,10,11,12,13,14]. The symptoms usually appear around the second to the third week of cancer treatment [15], and although they are mild at first, they can become severe over time [16]. The impact of cancer on patients is multifaceted and has been documented in some studies [17,18]. The emotional consequences observed in people with cancer are depression, worry, fear, anger and guilt [19,20]. Furthermore, emotional distress is often referred to as the “sixth vital sign” for cancer patients, and healthcare professionals routinely evaluate it in conjunction with other vital signs like pulse, breathing, blood pressure, temperature and pain [21]. Besides the direct impacts, the connections between cancer and psychosocial consequences can also be influenced by the physical symptoms of cancer or the adverse effects of its treatment [22], such as nausea, vomiting and pain. This premise is supported by some studies that show that higher degrees of pain from cancer are related to a decrease in social activities, support and functioning [23]. Simultaneously, fatigue, which is a prevalent symptom among cancer patients, has been identified as a predictor of depression and anxiety [24]. Therefore, the impact of body image change due to the cutaneous toxicity of cancer treatment on the emotional and psychological components of the patient remains to be studied. Accordingly, this review could serve to identify gaps regarding the psychosocial impact of dermatological side effects resulting from cancer treatment and propose future studies and clinical practices to minimize this impact through clinical and psychological therapeutic management strategies. The authors present a discussion of the state of the art and reflections on future research needs.

## 3. Methods

### 3.1. Search Strategy

The Preferred Reporting Items for Systematic Reviews and Meta-Analyses (PRISMA) guided a systematic search through the PubMed, Cochrane Library and PyscNet databases. The keywords used combined: “cancer” with “chemotherapy” or “treatment”, or “dermatological side effects” or “skin side effects” or “alopecia” or “body image” and “quality of life” or “psychosocial impact” or “social inhibition” or “cognitive fusion”. Some additional records were assessed through these articles’ references. The resulting studies were filtered. However, no time limit was imposed. All authors have been actively involved in all the other phases of the review process.

### 3.2. Inclusion Criteria

The studies’ titles and abstracts were assessed and screened for inclusion following specific criteria: (a) complete studies (no protocols); (b) studies correlating cancer and its psychological and psychosocial outcomes; (c) studies involving specific cancer treatments and their psychological and psychosocial outcomes; (d) studies assessing the psychological and psychosocial consequences of skin toxicities; and (e) studies published in peer-reviewed journals. The articles considered are in the English and French languages.

### 3.3. Exclusion Criteria

Studies were excluded if they only focused on cancer and its physical outcomes and not on its psychological or psychosocial impacts or if they focused only on the dermatological literature, not establishing an association with psychology. Studies that focused only on the impact of the family/loved ones/caregivers (as their perspective could be biased) were also excluded. Works that only mentioned pharmacological therapies and with a small sample size (<10 participants) were also excluded.

### 3.4. Screening

The title/abstract of the studies was selected independently by each of the authors, always based on the inclusion and exclusion criteria.

### 3.5. Quality Assessment

The studies’ methodological quality was assessed by its journal Q Index using Scimago Journal & Country Rank. Out of the seventeen selected articles, nine were in the top impactful 53% (Q1) [25,26,27,28,29,30,31,32,33], seven were in the less impactful 41% (Q2) [34,35,36,37,38,39,40], and one was in the lesser impactful 6% (Q3) [41]. To detail methodological issues and to make future investigation recommendations, studies lower than the ideal Q1 were still included.

## 4. Results

The initial searches through the databases considered resulted in a total of 725 articles, of which 213 (29.38%) were removed because they were duplicates. The abstracts of the remaining 512 (70.62%) studies were screened, and 397 (54.76%) were excluded. From the remaining 115 (15.86%) full-text articles, 98 (13.10%) were ruled out, resulting in 17 (2.76%) articles for review (Figure 1).

### 4.1. Studies’ Characteristics

The studies’ characteristics (*n* = 17) are reported in Table 1. One study (6%) includes various types of cancer patients [33], three studies (18%) narrow their samples by recruiting patients with a specific cancer type [26,34,38], nine studies (53%) address the impact of treatments on quality of life [25,26,27,28,32,33,34,37,40] and one study (6%) addresses the role on body image [39]. There are four literature review studies [31,35,39,41], two retrospective studies [25,27], three quantitative studies [29,34,37], one multi-center randomized study [30], one prospective longitudinal study [36] and one prospective study [38].

### 4.2. Studies’ Results Summary

The analysis of Table 1, Table 2 and Table 3 allows for the understanding that while the methodology and instruments may differ (Table 2 and Table 3), most studies are highly qualified in terms of metrics (Table 1), mostly Q1 and Q2, which demonstrate how valued studies on this topic are. The most studied variables were sociodemographic and clinical characteristics, such as age, sex, type of cancer and treatment (Table 2). QoL is clearly the focus of most articles, underlining aspects related to the psychological well-being or social interactions of the patients. The side effects of cancer treatment and medication are extensively addressed in the reviewed studies, demonstrating the importance given to this specific factor in the well-being of patients. Treatment with EGFRI seems to result in significant physical and psychosocial discomfort of patients and was associated with more dermatological side effects, therefore highlighting the need to further invest in the well-being of patients during the treatment phase. The sense of hope and despair during the illness has been described in some studies, once more within the contest of QoL. While dealing with chemotherapy, there is a perceived need to move away from the disease symptoms and focus on the QoL. These connections need to be better explored, tailored to each patient beyond the improved aspects of health and also specifically address the psychological aspects involved in the QoL.

Thus, skin changes are frequently perceived as secondary issues in the context of cancer, often overshadowed by more evident symptoms of the disease. Nevertheless, these dermatological changes can exert a substantial and sometimes underestimated impact on a patient’s QoL, primarily due to their unpredictable nature. It is well established that healthcare practitioners frequently possess limited knowledge regarding these dermatological effects, thereby impeding their ability to provide comprehensive support not only to patients but also to their families. Consequently, it becomes imperative for healthcare professionals to proactively engage in self-education concerning the potential side effects of cancer treatments, particularly those related to changes in physical appearance that may evolve gradually over time. Furthermore, healthcare providers should endeavor to facilitate the holistic management of patients, addressing both the physical and psychological repercussions of these dermatological side effects. This not only empowers patients with a deeper understanding of what they might encounter during their cancer journey but also equips healthcare professionals with the necessary tools to offer more comprehensive and empathetic care. In doing so, the aim is to enhance the overall well-being and QoL of cancer patients and their families, acknowledging the significant role that skin changes can play in their cancer experience.

Clearly, through this systematic review, the two research questions being addressed in this study were answered: The dermatological side effects of the cancer treatment result in psychosocially impacting the patients (RQ1). The psychosocial impact of the dermatological toxicities of the cancer treatment manifest in patients’ lives is demonstrated in the reviewed studies (RQ2).

Table 3 details the instruments used in the analyzed studies.

## 5. Discussion and Conclusions

Patients with the most varied types of cancer survive longer and mention an improvement in the level of QoL as a consequence of new cancer therapies. Dermatological changes are a common side effect of this treatment [12,42]. They can range from mild to severe and can affect a patient’s physical appearance, comfort and QoL. Some of the most common skin alterations in cancer patients include dryness, itching, rashes, changes in skin color, hair loss or nail changes. These skin effects can be caused by a variety of factors, including chemotherapy, radiation therapy, immunotherapy and surgery. The severity of skin changes can vary depending on the type of cancer treatment, the patient’s individual response to treatment and the patient’s skin type. In some cases, skin changes can be temporary and improve after treatment is finished [43,44]. However, in other cases, these cutaneous alterations can be permanent. It is important for cancer patients to be aware of the potential skin changes that can occur during treatment and the effect they may have on their QoL. Skin toxicities are often understood as minor concerns when compared to cancer symptoms, but they can seriously affect patients’ QoL due to their unforeseen nature. They can impact a patient’s self-esteem, body image and sense of well-being. Skin changes can also make it difficult for patients to perform everyday activities, such as bathing, dressing and sleeping [25,26]. Few studies have addressed the psychosocial impact of the side effects of cancer treatment, as demonstrated in this review. However, the psychological and social impact of dermatoses is well documented, and so it is expected that this impact will be high, similar to what happens with these diseases [45,46,47,48]. Health professionals often have limited knowledge about these effects, which limits their ability to effectively support not only patients but also their families. Therefore, this impact is not considered in therapeutic strategies, and if it were, it would greatly benefit the patient [49]. It is important for healthcare providers to be aware of the potential skin alterations that can occur in cancer patients. They should also be able to provide patients with information and support to help them manage these effects. It is important that these professionals educate themselves and the patients about the possible side effects of cancer treatment, such as modifications in the appearance that may manifest over time, and also seek to help the management of patients’ physical and psychological dimensions.

Following the PRISMA guidelines, this article aimed to address the psychosocial impacts of cancer treatment-related dermatological side effects and how they can be synthesized through a systematic review. While there is a recognition of the adverse impact of dermatological side effects caused by cancer treatments on patients, there remains a dearth of understanding regarding how coping with these effects influences the psychological and social dimensions of cancer patients’ daily lives. This review highlights that the majority of the studies developed have focused on the physical aspects of cancer, and no additional relevance was given to the dermatological aspects associated with the side effects of the treatment. Additionally, research studies concentrating on the consequences of dermatological alterations as their primary focus did not establish a direct connection to cancer itself. This review aimed to contribute to further clarification in relation to the aspects contributing to clarify the key points in the treatment of skin cancer diseases, adding knowledge in this respect. The development of further research studies that are able to assess the psychosocial impact of the dermatological side effects of cancer treatment and psychological interventions to minimize this impact will allow a significant improvement in healthcare for patients undergoing cancer treatment.

## Figures and Tables

**Figure 1 healthcare-11-02621-f001:**
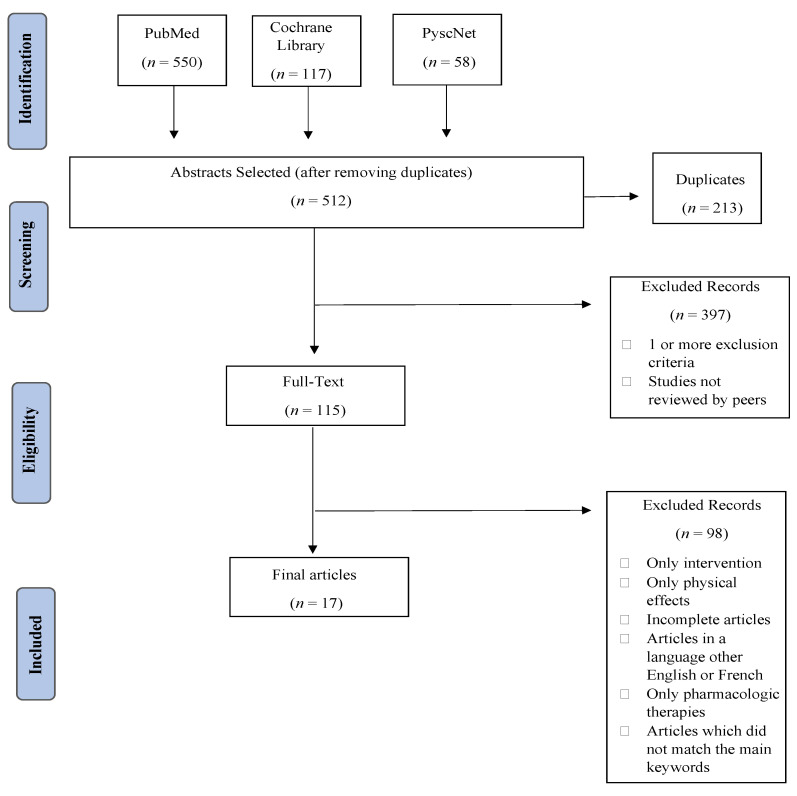
PRISMA flow diagram comprising the different phases of this systematic review.

**Table 1 healthcare-11-02621-t001:** Resume of the studies reviewed for this literature review.

Study Title	Reference and Metrics ^1^	Main Aim	Methodology	Main Results	Discussion/Conclusions
**Effects of epidermal growth factor receptor inhibitor-induced dermatologic toxicities on quality of life**	Joshi et al. [25]; Q1 (high)	Examine the effect of skin toxicities caused by epidermal growth factor receptor (EGFR) inhibitors on QoL.	Exploratory study	Consistent with a higher severity of rash grade, there was a rise in median scores for symptoms, emotions, functioning and overall scores. Conversely, there was a negative correlation between age and emotions as well as the overall score. A noteworthy disparity emerged between patients aged ≤50 years and those aged >50 years in terms of symptoms, emotions, functioning and overall scores. However, no significant differences were observed in terms of QoL concerning skin patch testing (SPT), gender, treatment modality or cancer type.	Toxicity, encompassing issues such as rash, xerosis, paronychia and pruritus, had an adverse impact on the QoL, with rash exhibiting the most substantial QoL reduction. Interestingly, younger patients reported lower overall QoL compared to their older counterparts who experienced similar toxicities. These findings underscore the utility of the NCI-CTCAE as a valuable tool for assessing the influence of rash on dermatology-specific QoL.
**Management of cutaneous side-effects of cetuximab therapy in patients with metastatic colorectal cancer**	Ocvirk et al. [26]; Q1 (high)	Achievements of a literature review on the management of skin toxicity during treatment with cetuximab.	Longitudinal study	Thirty-one patients with metastatic colorectal cancer treated with cetuximab were examined. A total of 6 patients had grade I rash, 16 had grade II, and 9 patients had type I acne as a rash. To a lesser extent, cases of itching, dry skin, scaling, capillary modification, conjunctivitis, telangiectasias, paronychias or fissures were observed. Cetuximab therapy was discontinued at grade III.	When using EGFRI treatment, it is crucial to identify and address adverse reactions to ensure the patient’s QoL and enable the uninterrupted continuation of therapy, avoiding the necessity of reducing or discontinuing the medication.
**Psychological effects of cetuximab-induced cutaneous rash in advanced colorectal cancer patients**	Romito et al. [34]; Q2 (medium)	Investigate the psychological and social sequelae of skin rash.	Exploratory study	A total of eighty individuals who had advanced colorectal cancer (CRC) and underwent treatment involving cetuximab were included in the study. Among these patients, 41% displayed signs of psychological distress. Concerning social avoidance, 53% of the participants reported that they did not avoid going out at all. The remaining respondents indicated that they either “strongly” (22%) or “moderately” (25%) avoided going out. Notably, the study found that psychological distress and social avoidance were not associated with the presence of skin rash but were instead linked to the patient’s QoL.	The presence of a skin rash did not appear to influence the psychological well-being or social interactions of the patients. This observation can be attributed to two plausible reasons: firstly, individuals with advanced cancer may view a skin rash as an inherent component of the multifaceted challenges posed by the disease itself; secondly, oncologists often motivate patients to persevere with their treatment regimens since a skin rash typically signifies a positive response to the therapy. Such expectations can instill hope and assist patients in coping with the side effects associated with the medication.
**Cutaneous side-effects in patients on long-term treatment with epidermal growth factor receptor inhibitors**	Osio et al. [27]; Q1 (high)	Conduct a prospective study in patients undergoing epidermal growth factor receptor inhibitors for more than 6 months.	Exploratory study	In total, 100% had cutaneous side effects at the time of the examination; 5% grade I or II folliculitis; 100% xerosis; 69% mucositis; 5% capillary abnormalities; 5% trichomegaly of the lashes; 56% facial hypertrichosis; 56% painful paronychia; 44% onycholysis; 5% needed a reduction dose or discontinuation EGFRI; 25% suffered moderate to strong impact on QoL.	Patients treated for cancer for longer periods can present chronic side effects. Cutaneous side effects were found in all patients treated with EGFRI for more than 6 months with a significant impact QoL, marked by significant physical and psychosocial discomfort. The clinical spectrum of skin manifestation varies over time.
**Patient perceptions of the side effects of chemotherapy: the influence of 5HT3 antagonists**	De Boer-Dennert et al. [28]; Q1 (high)	Assess the influence of 5HT3 antagonists on how patients perceive the adverse effects of chemotherapy.	Exploratory study	Patients ranked nausea, hair loss and vomiting as the most distressing side effects, as they can impact QoL and compliance with treatment. The ranking of the four most distressing side effects was quite consistent for gender, age and marital status. In comparison to men, women gave greater importance to hair loss over vomiting, and they also rated emotional distress (depression), anxiety or tension as more significant concerns than men did. On the other hand, men expressed more apprehension regarding treatment attendance, the duration of clinic visits and infertility issues. Notably, infertility was a source of greater distress among younger patients. As patients grew older, the importance of factors like the impact on family and partners and feelings of anxiety or tension diminished, while concerns about constipation and the need for injections became more prominent. Additionally, the anticipation of undergoing treatment had a lesser effect on older patients.	The findings from the study emphasize the need to stay attuned to how patients perceive the side effects of chemotherapy, which might not align with the perspectives of healthcare professionals. While acknowledging the significance of introducing new and effective supportive care measures like 5HT3 antagonists, the study also urges caution against overly optimistic interpretations of their impact.
**On the receiving end. V: Patient perceptions of the side effects of cancer chemotherapy in 1993**	Griffin et al. [29]; Q1 (high)	Determine and prioritize the symptoms encountered by patients undergoing cancer chemotherapy.	Longitudinal study	In 1983, a study highlighted vomiting and nausea as the predominant symptoms among patients. Given advancements in antiemetic treatments and shifts in cancer chemotherapy approaches, it was expected that changes might have occurred in how patients perceived their symptoms. In 1993, the study was repeated, involving 155 cancer patients undergoing chemotherapy at a large urban teaching hospital. Patients were asked to select symptoms from a list and identify the five most troublesome ones. On average, patients reported experiencing 20 symptoms (13 physical and 7 psychosocial). Nausea emerged as the most distressing symptom, followed by fatigue and hair loss. Interestingly, vomiting, previously the most severe symptom in 1983, now ranked fifth. Variations in the symptoms experienced and deemed most severe were observed among different chemotherapy regimens, across age groups, and between male and female patients.	The findings indicate a decrease in the intensity of certain symptoms encountered during chemotherapy and a transition from a focus on physical symptoms to a greater emphasis on psychosocial concerns.
**Effect of Peri-operative Chemotherapy on the QoL of Patients with Early Breast Cancer**	Kiebert et al. [30]; Q1 (high)	Examine how perioperative chemotherapy affects the physical, psychological and social well-being, as well as the activity level, of individuals diagnosed with early-stage breast cancer.	Exploratory study	Out of the participants, 24 women underwent perioperative chemotherapy, while 29 did not receive this treatment. During the initial 2 months following surgery, patients who received perioperative chemotherapy did not report a higher overall occurrence of physical symptoms compared to the control group. The impact on body image, fear of recurrence and fear of death using six questions, the reliability of which had been previously established in a study involving breast cancer patients, were assessed. Notably, there was a noteworthy difference between the two groups in terms of fatigue, with the perioperative chemotherapy group reporting more fatigue than the control group. Additionally, complete hair loss was more commonly reported by patients who underwent perioperative chemotherapy. However, the subjective evaluation of physical well-being among patients who received perioperative chemotherapy did not significantly diverge from that of the control group. Moreover, there were no discernible disparities in psychological well-being, concerns and fears, daily activity performance or the overall assessment of life between the two groups.	Prior to this research, the prevailing assumption was that chemotherapy-induced alopecia would inevitably have a detrimental impact on the QoL. However, the findings revealed that the interconnections between these factors were more intricate than previously believed. The question that remains unresolved is how breast cancer and/or its surgical interventions shape the perception of alopecia as a side effect of adjuvant chemotherapy.
**Chemotherapy-induced alopecia: psychosocial impact and therapeutic approaches**	Hesketh et al. [35]; Q2 (medium)	Identify the psychosocial effects resulting from chemotherapy-induced alopecia.	Literature review	Chemotherapy-induced alopecia (CIA) could affect QoL and lead to significant levels of anxiety, depression, negative body image, low self-esteem and reduced sense of well-being.	The CIA approach should be tailored to each patient’s specific requirements, with a particular emphasis on addressing the precise timing of hair loss. Both support groups and self-care strategies constitute essential elements of any comprehensive management approach.
**Changes in self-concept and body image during alopecia induced cancer chemotherapy**	Münstedt et al. [36]; Q2 (medium)	Investigate how chemotherapy-induced hair loss affects different aspects of body image perception.	Longitudinal study	Upon histological confirmation predominantly indicating ovarian cancer, a group of 29 patients who received a chemotherapy regimen known to induce complete alopecia (loss of hair) was analyzed. The assessment was conducted prior to the initiation of treatment, then repeated once hair loss was complete, and finally after the completion of therapy when patients had already undergone hair regrowth. Across all scales, the results demonstrated a deterioration during chemotherapy, but they did not return to baseline or improve when patients experienced hair regrowth. The findings indicated that 73.3% of the patients did not feel as self-confident as they did before undergoing treatment, and for 46.6%, alopecia represented the most distressing side effect of chemotherapy.	Given the absence of any chemotherapeutic regimen or alternative treatment capable of averting alopecia, one of the following conclusions can be drawn: The observed differences may not solely be attributed to alopecia but could also result from coping mechanisms triggered by chemotherapy, potentially exacerbated by the presence of alopecia. Alternatively, these changes persist even after chemotherapy has ceased. The regrowth of hair and other adaptive processes do not lead to the restoration or enhancement of the compromised body image and self-concept.
**Impact of Skin Toxicities Associated with Targeted Cancer Therapies on Body Image: A Prospective Study**	Charles et al. [37]; Q2 (medium)	Describe the changes in body image that occur due to skin toxicity and their psychosocial impact on patients.	Exploratory study	In total, 94% developed skin toxicity. Body satisfaction remained stable and even slightly better during this period; 1/3 of the participants reported body image problems. The initial levels of body satisfaction and depression seemed to have a significant connection with the emergence of body image issues following three months of treatment.	Regarding the management of dermatological aspects, there appeared to be no apparent correlation between skin toxicities and body image issues. Nonetheless, it is essential for physicians to recognize that factors like body satisfaction and depressive symptoms present at the start of therapy play a pivotal role and should be taken into account to prevent the deterioration of body image and overall QoL.
**Cytokines, Fatigue, and Cutaneous Erythema in early stage Breast Cancer Patients Receiving Adjuvant Radiation Therapy**	De Sanctis et al. [38]; Q2 (medium)	Investigate the possible association of the development of high-grade erythema of the breast skin during radiation treatment with fatigue.	Longitudinal study.	Among the 40 patients who underwent management, assessments were conducted before, after radiotherapy (at 4 weeks), and during follow-up (6 months post-radiotherapy). During these evaluations, symptoms of fatigue, skin erythema and levels of circulating proinflammatory cytokines were recorded. Among these patients, 17.5% experienced fatigue without concurrent depression or anxiety, and grade ≥2 erythema was observed in 5 out of these 7 patients. The blood markers demonstrated a notable impact on fatigue. Interestingly, there appeared to be an apparent rise in fatigue, erythema and proinflammatory markers between the fourth and fifth weeks of treatment, followed by a subsequent decrease after radiotherapy.	The research findings imply that fatigue is linked to the presence of severe breast skin erythema during radiotherapy, likely due to elevated cytokine levels. These increased cytokine levels were found to be associated with concurrent high-grade breast skin erythema, potentially contributing to the biological mechanisms underlying fatigue. This suggests the possibility of developing radiation therapy modifications or novel medications specifically targeting erythema, which could reduce the intensity of skin erythema and fatigue. Such interventions could lead to improved therapy adherence and enhanced QoL for patients.
**Body image issues in women with breast cancer**	Helms et al. [39]; Q2 (medium)	Investigate concerns related to body image and assess psychological adaptation among women diagnosed with breast cancer.	Literature review	Women who have been diagnosed with and treated for breast cancer encounter various physical transformations. These changes encompass both potentially life-threatening alterations and others that could be characterized as primarily aesthetic. Among the effects, women with breast cancer may experience weight gain, hair loss and significant changes in breast appearance. These physical transformations have the potential to influence a woman’s overall sense of well-being and her ability to adapt to life following cancer treatment. Moreover, it appears that a strong emphasis on body image may amplify the psychological impact of these cosmetic changes.	While there existed a body of research investigating body image concerns among women with breast cancer that could be built, there were a number of clear areas for future investigation, such as better methodology, etiology of weight gain in relation to breast cancer, limited psychosocial oncology research and practice, lack of discussion in body image research related to breast cancer and also lack of knowledge in the impact of real physical changes on psychological well-being.
**Dermatologic side effects associated with the epidermal growth factor receptor inhibitors**	Agero et al. [31]; Q1 (high)	Describe the clinical characteristics of the dermatological adverse reactions caused by EGFR and discuss the pathology, possible causes and suggested treatments.	Literature review	The most common adverse dermatological effect was mild skin toxicity, characterized by a follicular and sterile pustular skin rash. In turn, the secondary adverse skin reactions observed include xerosis, pruritus, paronychia, capillary abnormality, stomatitis/mucositis, hypersensitivity reactions and nail changes.	Although the precise mechanism for the development of rash is not well defined, it was related to EGFRI signaling pathways in the skin and can serve as a visible marker of antitumor activity and therapeutic efficacy. These dermatologic reactions were common and generally mild but can cause clinical distress to the patient. It is important that dermatologists can diagnose these side effects and differentiate them from other skin disorders.
**Management of skin adverse reactions in oncology**	Silva et al. [41]; Q3 (low)	Summarize information pertaining to the prevention and treatment of skin toxicity resulting from chemotherapy and targeted cancer therapies.	Systematic review	Emphasizing the significance of patient education, three fundamental actions are underscored: cleansing, skincare and protection. This comprehensive approach is crucial for maintaining healthy skin, especially in cases where the skin has been compromised due to oncology treatments. The management of skin-related adverse reactions resulting from cancer treatments has been notably diverse, mainly due to the limited availability of well-founded, evidence-based treatments. Among the various adverse effects studied, papulopustular eruption, xerosis and hand–foot syndrome have received the most attention. Notably, the prevention of xerosis stands out as the approach with the strongest support from level II studies. Concerning treatment, the use of antibiotics to address papulopustular eruption caused by anti-epidermal growth factor receptor agents represents the most evidence-based strategy. In general, the number of studies categorized in the literature with a level II evidence rating (52%) is similar to those classified as level IV (33%).	Skin toxicity frequently occurs during combined cancer treatments and can lead to pain, discomfort, irritation, itching and even treatment delays or interruptions. It is crucial to educate patients about the potential risks and causes of skin toxicities before initiating anticancer therapy. Early consideration of skin adverse reaction management is essential to uphold the QoL for patients undergoing cancer therapies.
**Effect of skin reactions on QoL for elderly women with breast cancer receiving chemotherapy**	Abd EL-rafea et al. [40]	Evaluate the impact of skin reactions on the QoL of elderly women undergoing chemotherapy for breast cancer.	Descriptive design	In the group of elderly women under investigation, 40% experienced an extremely profound impact on their QoL due to skin symptoms. Furthermore, 64% of them reported an extremely severe effect on their emotions and functioning, resulting in a total mean QoL score of 103.72 ± 11.6. Additionally, statistically significant associations were identified between the QoL and factors such as age, lack of formal education, insufficient monthly income, the presence of chronic diseases, the stage at which breast cancer was diagnosed, and the receipt of ten or more chemotherapy cycles.	Regarding the overall aspects of QoL, the research findings indicated that approximately two-thirds of the elderly women included in the study reported an extremely significant impact of their skin symptoms on their emotional well-being and daily functioning. Furthermore, nearly half of these individuals stated that their skin symptoms had an extremely profound effect on their overall QoL.
**Assessment of QoL and Treatment Outcomes of Patients With Persistent Postchemotherapy Alopecia**	Freites-Martinez et al. [32]; Q1 (high)	Describe the clinical manifestation of patients who experience persistent chemotherapy-induced alopecia (pCIA) or endocrine therapy-induced alopecia following chemotherapy (EIAC) and evaluate their QoL and treatment outcomes.	Retrospective multi-center cohort	A total of 98 women diagnosed with pCIA (median age, 56.5 years, ranging from 18 to 83 years) and 94 women with EIAC (median age, 56 years, ranging from 29 to 84 years) were included in the study. In the case of pCIA, taxanes were the most common agents associated with the condition for 80 patients (82%), while aromatase inhibitors were the most common agents linked to EIAC for 58 patients (62%). In terms of the clinical presentation, diffuse alopecia was more prevalent in patients with pCIA compared to those with EIAC (observed in 31 out of 75 (41%) vs. 23 out of 92 (25%); *p* = 0.04), and the severity of alopecia, as per Common Terminology Criteria for Adverse Events, version 4.0, grade 2, was notably higher among patients with pCIA (29 out of 75 (39%) vs. 12 out of 92 (13%); *p* < 0.001). Both groups reported experiencing a negative emotional impact. Following treatment with topical minoxidil or spironolactone, a moderate to significant improvement was observed in 36 out of 54 patients with pCIA (67%) and 32 out of 42 patients with EIAC (76%).	Cytotoxic chemotherapy treatments have the potential to cause lasting harm or depletion of epithelial hair follicle stem cells, which are essential for the regeneration of hair follicles. This harm may elucidate the notably higher prevalence of diffuse alopecia observed in patients with pCIA compared to those with EIAC. In addition to the scalp, it was observed that eyebrow and eyelash alopecia were present in 37% of patients with pCIA, contributing to an adverse psychosocial impact, as indicated in the QoL analyses. Studies have shown that scalp cooling can prevent severe alopecia in 51% of breast cancer patients. Moreover, among patients with pCIA, there was a generally moderate to significant clinical improvement observed in 36 out of 54 individuals (67%) who received treatment with topical minoxidil, 5% and/or oral spironolactone. Multivariate analysis demonstrated a negative emotional impact in both groups when compared to other domains, such as symptoms and functioning.
**The Impact of Skin Problems on the QoL in Patients Treated with Anticancer Agents: A Cross-Sectional Study**	Lee et al. [33]; Q1 (high)	Evaluate the impact of anticancer agents on patients’ QoL	Cross-sectional study	Out of the 375 participants who underwent anticancer therapy, 136 (36.27%) received treatment for breast cancer, and 114 (30.40%) were treated for colorectal cancer. The analysis revealed that factors associated with greater dermatology-specific QoL disruption included being female, having breast cancer, using targeted agents and undergoing longer durations of anticancer therapy. Furthermore, specific dermatological symptoms such as itching, dry skin, easy bruising, pigmentation issues, papulopustules on the face, periungual inflammation, nail abnormalities and palmoplantar lesions were linked to significantly higher scores on the Dermatology Life Quality Index (DLQI). Among these symptoms, periungual inflammation and palmoplantar lesions were associated with the highest DLQI scores.	The findings of this study indicated that factors such as being female, having breast cancer, using targeted agents and undergoing extended periods of anticancer therapy were linked to increased disruption in dermatology-specific QoL. All of the skin issues examined, with the exception of hair loss, had a notable impact on dermatology-specific QoL. Notably, periungual inflammation and palmoplantar lesions were associated with the highest DLQI scores. These results could serve as valuable insights for clinicians when counseling and managing patients undergoing anticancer therapy.

^1^ According to Scimago Journal & Country Rank. Acronyms: EGFRI: Epidermal Growth Factor Receptor Inhibitor; MKIs: multikinase inhibitors; DLQI: Dermatology Life Quality Index; RCTs: randomized controlled trials.

**Table 2 healthcare-11-02621-t002:** Variables, analysis and statistical methods used in the studies reviewed for this literature review.

Reference	Analysis and Statistical Methods	Variables
Joshi et al. [25]	*t* tests; Bonferroni method; Spearman correlation; Wilcoxon rank-sum tests; Kruskal–Wallis test	Sex; age; EGFRI; EGFRI-related reactions; Fitzpatrick skin phototype; quality of life (QoL)
Ocvirk et al. [26]	Not discriminated	Gender; age; rash grade; hair modifications; fissures; paronychia
Romito et al. [34]	Chi-Square test; Pearson’s correlation; Spearman’s test; *t* test; SPSS	Sex; age; colorectal cancer; social avoidance; psychological distress; skin rash; QoL
Osio et al. [27]	Not discriminated	Sex; age; EGFRI-related reactions; associated chemotherapy; duration of EGFR (months); QoL
De Boer-Dennert et al. [28]	Chi-square test; Fisher’s test	Sex; age; marital status; tumour types; chemotherapy
Griffin et al. [29]	LASA methodology	Sex; age; months from first diagnosis; marital status; type of cancer; extent of disease; intent of therapy; patient status; antiemetic regimens
Kiebert et al. [30]	*t* tests and *X*^2^ tests; two-way analyses of variance	Age; stage of disease; type of surgery; breast cancer; radiotherapy; adjuvant chemotherapy; menopausal status; time since surgery; QoL
Münstedt et al. [36]	MANOVA; *t* tests; Bonferroni method; Cronbach’s Alpha; SPSS	Age; gynaecological malignancy; mainly ovarian cancer; chemotherapy
Charles et al. [37]	Descriptive statistics; McNemar and Wilcoxon tests; Chi-square test; Mann–Whitney *U* test; Spearman’s coefficient; SPSS	Age; sex; marital status; cancer site; treatment; included in clinical trial; previous dermatological symptoms
De Sanctis et al. [38]	Student’s *t* test (Heckman two-step correction)	Age; early-stage breast cancer patients who underwent conservative surgery and radiotherapy; conserving surgery; histological type; tumor grade
Abd EL-rafea et al. [40]	Media; correlations; multiple linear regression model	Age; marital status; educational level; living condition; responsible for women’s care; monthly income; suffering from chronic diseases; duration of breast cancer diagnosis; stages of breast cancer at diagnosis; time since receiving chemotherapy; number of total chemotherapy cycles; previous chemotherapy history
Freites-Martinez, et al. [32]	Descriptive statistics; univariate regression tests; multivariate logistic regression	Alopecia grading and pattern; trichoscopy; response to therapy; QoL
Lee et al. [33]	Student’s *t* test; Jonckheere–Terpstra test; correlations	Sex; age; current chemotherapy; duration of current chemotherapy; radiotherapy history; presence of skin problems; QoL.

Note: The articles where references are Hesketh et al. [35], Helms et al. [39], Agero et al. [31], and Silva, D. et al. [41] are literature reviews or systematic reviews; therefore, they do not have the information required in this table.

**Table 3 healthcare-11-02621-t003:** Instruments used in the studies reviewed for this review.

Reference	Instruments
Joshi et al. [25]	The National Cancer Institute Common Terminology Criteria for Adverse Events (NCI-CTCAE)—version 3.0 Skindex-16 Fitzpatrick SPT
Ocvirk et al. [26]	NCI CTCAE, v3.0
Romito et al. [34]	The Functional Assessment of Cancer Therapy—Colorectal (FACT-C) The Psychological Distress Inventory (PDI) The National Cancer Institute Common Terminology Criteria for Adverse Events (NCI-CTCAE)
Osio et al. [27]	Questionnaire Dermatology Life Quality Index (DLQI) Common Terminology Criteria for Adverse Events (NCI-CTCAE)
De Boer-Dennert et al. [28]	Coates et al. (1983) Questionnaire (perception of the side effects of cancer chemotherapy)
Griffin et al. [29]	Coates et al. (1983) Questionnaire (perception of the side effects of cancer chemotherapy) Cancer Linear Analogue Self-Assessment Scales (CLASA)
Kiebert et al. [30]	Rotterdam Symptom Checklist (RSCL) Adapted Revenson Scale (and others) Modified EORTC Quality of Life Study Group questionnaire
Münstedt et al. [36]	Frankfurt Body-concept Scale (FKKS) Frankfurt Self-concept Scale (FSKN)
Charles et al. [37]	The National Cancer Institute Common Terminology Criteria for Adverse Events version 4.0. (NCI-CTCAE) Body Image Questionnaire (BIQ) Physical Attitudes Questionnaire (PAQ) Beck Depression Inventory-II (BDI-II)
De Sanctis et al. [38]	Cancer Therapy Fatigue subscale (FACT-F)—a component of the FACT-G quality of life questionnaire Hospital anxiety and depression scale (HADS)
Abd EL-rafea et al. [40]	Structured interview questionnaire; Skindex-29
Freites-Martinez, et al. [32]	Hairdex questionnaire; 4-point scale
Lee et al. [33]	Dermatology Life Quality Index (DLQI); Questionnaire survey; Patirent’s medical records review.

Note: The articles where references are Hesketh et al. [35], Helms et al. [39], Agero et al. [31], and Silva, D. et al. [41] are literature reviews or systematic reviews; therefore, they do not have the information required in this table.

## Data Availability

All data used are included in the study.

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
