# Peer review of "Dermatological Side Effects of Cancer Treatment: Psychosocial Implications—A Systematic Review of the Literature"

_healthcare, 2023, doi:10.3390/healthcare11192621_

Round 1
Reviewer 1 Report
This study aims to explore and synthesize the psychosocial impact of dermatological side effects of cancer treatment. It is relevant and valuable for healthcare providers because it adds evidence to support their practice.
Reviews and Meta-Analyses (PRISMA) guidelines were followed, but I have identified some aspects that could be deepened or integrated to increase the quality of the article.
The title should identify the study as a systematic review. It is a systematic review rather than a review of the literature.
The research aim is precise but does not provide an explicit statement of the question (s) of the study.
The methods need to be further specified to allow the study to be replicated:
- P.2, lines 18-20 – Search strategy isn´t clearly described. It is written that “the keywords were used in a combined manner”, but this is unclear to me. The keywords cancer treatments were not always in all combinations.
- P.3, lines 33-34 It is written: “Works that only mentioned pharmacological therapies and with a small sample size were also excluded.” To better understand the exclusion criteria, it would be helpful to identify the small sample size.
- P.3, line 36 – It is written that the studies were selected independently by each author, but the PRISMA checklist guides this to be done at all phases of the review.
- P.4, figure 1. Two or more exclusion criteria were considered in the exclusion of 397 studies, but in the methods section, it was determined that one exclusion criterion would be sufficient for the excluding studies.
- Table 1. I do not understand how, in a study where the title and objective specify the psychosocial impact of dermatological side effects of cancer treatment, studies not related to cancer treatment are selected, such as those on psoriasis and bullying. I question again whether the keyword cancer treatments was absent in the combinations.
The characteristics and results of the studies are clearly described, but I would like to see a data synthesis with answers to the stated objectives and research questions.
Author Response
The authors are grateful to the reviewer for the new important comments and suggestions that contribute to significantly improve the manuscript.
The authors provide a detailed answer to each of the reviewer comments.
Additional changes were now highlighted in YELLOW in the manuscript.
Reviews and Meta-Analyses (PRISMA) guidelines were followed, but I have identified some aspects that could be deepened or integrated to increase the quality of the article.
The title should identify the study as a systematic review. It is a systematic review rather than a review of the literature.
Response1: The title was changed in accordance to the suggestion.
The research aim is precise but does not provide an explicit statement of the question (s) of the study.
Response 2: This manuscript intends to identify, in a revision of the literature, the following questions:
- i) Do the dermatological side effects of the cancer treatment present any psychosocial impact for patients?
- ii) How does the psychosocial impact of the dermatological toxicities of the cancer treatment manifest in patients’ life?
These questions were added to the introduction of the paper.
The methods need to be further specified to allow the study to be replicated:
- P.2, lines 18-20 – Search strategy isn´t clearly described. It is written that “the keywords were used in a combined manner”, but this is unclear to me. The keywords cancer treatments were not always in all combinations.
Response 3: We appreciate this comment and to clarify the search strategy used the order and way in which the keywords were associated were replaced in the text.
- P.3, lines 33-34 It is written: “Works that only mentioned pharmacological therapies and with a small sample size were also excluded.” To better understand the exclusion criteria, it would be helpful to identify the small sample size.
Response 4: The number of participants of the exclusion criteria were now identified in the main text.
- P.3, line 36 – It is written that the studies were selected independently by each author, but the PRISMA checklist guides this to be done at all phases of the review.
Response 5: This sentence only relates to the article selection item; however all methodological phases of the systematic review process involved the participation of all authors. We have now clarified this in the manuscript.
- P.4, figure 1. Two or more exclusion criteria were considered in the exclusion of 397 studies, but in the methods section, it was determined that one exclusion criterion would be sufficient for the excluding studies.
Response 6: We appreciate the analysis, the figure 1 was corrected, it was a mistake.
- Table 1. I do not understand how, in a study where the title and objective specify the psychosocial impact of dermatological side effects of cancer treatment, studies not related to cancer treatment are selected, such as those on psoriasis and bullying. I question again whether the keyword cancer treatments was absent in the combinations.
Response 7: In agreement with the comment, table 1 was revised and some articles were now removed.
The characteristics and results of the studies are clearly described, but I would like to see a data synthesis with answers to the stated objectives and research questions.
Response 8: Aiming to address the reviewer’s comments, the manuscript now includes new content on this.

Reviewer 2 Report
The authors wanted to provided a review with the aim to synthesize and discuss the psychosocial impact of dermatological effects of cancer treatment and assess whether the phycological and emotional impact is integrated into the therapeutic strategies of the treatment of cancer patients.
It is a very interesting and important topic, but I think the authors didn't hit the target. They reported a great number of studies and they resume them in a table but they did not give a lecture and a discussion of the results. They just collected the discussion of each study. Moreover studies reported with references from 74 to 77 are not about cancer patients, so they should be removed, otherwise they could be included as background knowledge about the importance of skin diseases on psychology regardless of being affected by a cancer.
In conclusion the authors should exclude references 74-77 from the review and they should add a "discussion section" in order to analyse and discuss the psychosocial impact of dermatological effects of cancer treatment and if it is integrated into the therapeutic strategies.
Author Response
The authors are grateful to the reviewer for the new important comments and suggestions that contribute to significantly improve the manuscript.
The authors provide a detailed answer to each of the reviewer comments.
Additional changes were now highlighted in YELLOW in the manuscript.
It is a very interesting and important topic, but I think the authors didn't hit the target. They reported a great number of studies and they resume them in a table but they did not give a lecture and a discussion of the results. They just collected the discussion of each study. Moreover studies reported with references from 74 to 77 are not about cancer patients, so they should be removed, otherwise they could be included as background knowledge about the importance of skin diseases on psychology regardless of being affected by a cancer.
In conclusion the authors should exclude references 74-77 from the review and they should add a "discussion section" in order to analyse and discuss the psychosocial impact of dermatological effects of cancer treatment and if it is integrated into the therapeutic strategies.
Response: In agreement with the comment, table 1 was revised and some articles were now removed. A discussion section was added in association with the conclusion item of the document.

Round 2
Reviewer 2 Report
Thank you for the changes, the manuscript is now suitable for pubblication.